# Enhanced Performance Electrochemical Biosensor for Detection of Prostate Cancer Biomarker PCA3 Using Specific Aptamer

**Sarra Takita** [1,*]**, Alexei Nabok** [1]**, Anna Lishchuk** [2]**, Magdi H. Mussa** [1,3] **and David Smith** [4]

[1]  Material and Engineering Research Institute (MERI), Sheffield Hallam University, Sheffield S1 1WB, UK
[2]  Department of Chemistry, University of Sheffield, Brook Hill, Sheffield S3 7HF, UK
[3]  The Institute of Marine Engineering, Science and Technology, London SW1H 9JJ, UK
[4]  Biomolecular Sciences Research Centre (BMRC), Sheffield Hallam University, Sheffield S1 1WB, UK
[*]  Correspondence: st3805@exchange.shu.ac.uk; Tel.: +44-74-0473-1366

**Abstract:** In the quest for the development of accurate, reliable, and cost-effective biosensing technology for early diagnostics of prostate cancer, we describe here an electrochemical biosensor combining a simple transducing method of differential pulse voltammetry (DPV) with an RNA-based aptamer labelled with a methylene blue redox group acting as a highly specific bioreceptor to the prostate cancer biomarker PCA3. A series of DPV measurements on screen-printed gold electrodes is functionalised with a redox-labelled aptamer in solutions (either buffer or synthetic urine) containing PCA3 in a wide range of concentrations from 0.1 picomolar (pM) to 10 nanomolar (nM). In these measurements, the current peak values correlate with the concentration of PCA3 and yield a low detection limit (LDL) of 0.1 pM. Furthermore, the binding kinetics study revealed the high affinity of the aptamer to the target PCA3 with the affinity constants $K_D$ of about $3.0 \times 10^{-8}$ molar. In addition, the AFM study showed the increase in the molecular layer roughness caused by the binding of PCA3, which is a large RNA molecular fragment.

**Keywords:** prostate cancer; PCA3 biomarker; aptamer; electrochemical sensors; digital pulse voltammetry; binding kinetics; atomic force microscopy

## 1. Introduction

Prostate cancer (PCa) is one of the most prevalent types of cancer globally. Because the tumour tends to develop slowly and without symptoms, it is the second leading cause of death in men in developed countries, and its incidence and mortality are growing [1–4]. The existing prostate cancer early diagnostics is based on the evaluation of the concentration of serum prostate-specific antigen (PSA) in the blood (so-called PSA blood test) [5–7]. However, the PSA biomarker, being organ-specific but not tumour-specific, has a low positive predictive value of 30% [8,9]. Elevated PSA values, especially those between 4 and 10 ng/mL, are referred to as the diagnostic grey zone. The increased PSA levels might be linked with various benign illnesses, such as benign prostatic hyperplasia and prostatitis [10]. In case of high PSA level, the standard procedure in PCa diagnostics requires digital rectal examination (DRE), which in simple terms is a microbiopsy, magnetic resonance imaging, and imaging computed tomography scan [11]. The above techniques are invasive and time-consuming and require the use of qualified professionals. Such valuable diagnostic resources should not be wasted on cases of false-positive PSA tests [12]. Moreover, on rare occasions, normal blood PSA levels (less than 4 ng/mL) can be observed in people having prostate cancer (false-negative PSA tests), which may lead to a much more dangerous situation of late PCa diagnosis [12].

It is evident that the development of particular prostate cancer markers and early detection approaches are critical components in the fight against cancer. This can be achieved by implementing PCa screening based on accurate tests that provide valuable information while causing minimum discomfort and hazard. The testing should ideally

be noninvasive, low-cost, accurate, and sensitive to improve cancer patients' survival rates and quality of life. Such noninvasive testing is made possible by liquid assays for specific tissue biomarkers found in biological fluids, such as blood, urine, and saliva. Significant progress in this development, for example, finding reliable biomarkers for PCa, has been achieved. There were several PCa biomarkers discovered in recent years [13], including the prostate cancer gene 3 (PCA3), a long noncoding RNA that is only produced in prostate cancer tissues [14–17]. The combination of both PCA3 and PSA biomarkers has significantly enhanced the specificity of PCa diagnostics [16,18,19]. Such an approach has been successfully implemented in the Progensa® PCA3 test (Hologic Inc., Bedford, MA, USA), which was approved for use in the US private medical care system as the first molecular diagnostic test based on urine [20]. It employs RT-PCR to quantify the PCA3 score by measuring the PCA3 lncRNA/PSA mRNA ratio in the first urine collected after the digital rectal examination (DRE). The Progensa test substantially improved PCa early diagnostics and reduced unnecessary prostate biopsies [20,21]. However, this test is still expensive, time-consuming, and unavailable to many countries' general public.

The development of novel biosensing technologies for detecting the prostate cancer biomarker PCA3 is highly important nowadays. Many PCA3 biosensors have been developed recently, for example, colorimetric detection of PCA3 [22] by reverse transcription quantitative PCR (RT-qPCR) and fluorescence detection by RT-PCR with thiolated primers combined with unmodified gold nanoparticles [23,24]. Collecting molecules from urine onto magnetic particles coated with specific oligonucleotides, transcription-mediated amplification (TMA) was employed to identify the mRNAs linked with PCA3 [25]. Our research using the optical spectroscopic ellipsometry method in total internal reflection mode (TIRE) combined with PCA3-specific aptamers immobilised on gold-coated glass slides showed a high sensitivity for PCA3 [26]. Unfortunately, despite high sensitivity and specificity, these methods require expensive specialised equipment, specialised laboratories, and high-qualification personnel and are, therefore, not suitable for point-of-care medical diagnostics.

Electrochemical biosensors combined with nucleic-acid-based bioreceptors are particularly interesting in this respect. They combine the high selectivity of biomolecular identification with the high sensitivity, rapid response, and portability of screen-printed electrode biochips [27,28]. Furthermore, electrochemical detection allows for multiplex detection and uses simple and inexpensive equipment that people can operate without specialised training, which is crucial for clinical point-of-care diagnostics [29,30]. Miniature electrochemical sensors could be developed using low-cost screen-printed electrodes (SPE) [31,32], which could be functionalised with self-assembled monolayers (SAM) to provide a structured and orientated layer for different biosensing purposes [33]. Despite increased interest in developing biosensor platforms that target PCA3, there have only been a few reports on electrochemical biosensors that detect PCA3 [34–36]. Our recent research combined the electrochemical methods with redox-labelled aptamers specific to PCA3 [37–39]. The detection principles of such aptasensors have been proven using different types of aptamers (RNA and DNA based) labelled with different redox tags (ferrocene and methylene blue) and different electrochemical techniques, namely, cyclic voltammograms (CV) [37–39], differential pulse voltammograms (DPV) [39], and electrochemical impedance spectroscopy (EIS) [37,38]. The resulting values of LDL in the picomolar range are similar or better as compared with those obtained in other research groups [34–36] and, therefore, very promising for the detection of traces of PCA3 in regular urine samples. In terms of sensitivity, the best results so far were obtained with the EIS method, which showed the lowest LDL of 0.3 pM [37]. DPV, the preferred electrochemical analytical method for tracking inorganic and biological medical components due to its excellent sensitivity [40–42], was not explored in full capacity in our research. Therefore, this work focused on using DPV in combination with an RNA-based aptamer labelled with a methylene blue (MB) redox group. The amperometry kinetics study of the aptamer/PCA3 binding was carried out in order to evaluate the aptamer affinity to PCA3. In addition to the detection

of PCA3 in buffer solution, we attempted the use of synthetic urine as a step towards the testing of real samples of urine of prostate cancer patients. Atomic force microscopy (AFM) was used as a complementary method for visualising the aptamer/PCA3 binding process.

## 2. Materials and Methods

### 2.1. Reagents and Apparatus

Phosphate buffer saline (PBS), HEPES binding buffer (HBB) pH 7.2–7.6, sulfuric acid, and Tris (2-carboxyethyl) phosphine hydrochloride (TCEP, 99%) were purchased from Sigma-Aldrich (UK). Meanwhile, prostate-specific antigen (PSA) from human semen (99%) and bovine serum albumin (BSA) (98%) and Surine™ negative control (pH 6.9) was acquired from Merck Life Science Ltd. (Dorset, UK). All the reagents used were of analytical grade. The biological target, for example, a 277-nt fragment of lncRNA PCA3 having a molecular mass of 25,048.3 g/mol, and a scrambled PCA3 (negative control) were purchased from Eurofins Genomics (Germany). For biorecognition of PCA3, the CG3 aptamer shown schematically in Figure 1a with the sequence of nucleotides MB-5′-AGUUUUUGCGUGUGCCCUUUUUGUCCCC(CH2)$_3$-SH-3′ established in [43] was synthesised and functionalised with thiol at 3′ terminal and methylene blue (MB) at 5′ terminal with HPLC purity by Sigma Aldrich (Merck Life Science UK Limited., Dorset, UK). This sequence is in reduced form after adding TCEP; the redox functional group of MB was used as an electron mediator to provide distinctive electrochemical properties. A stock solution of the RNA aptamer (100 μM) was prepared in Tris buffer (10 mM Tris-HCL, 0.5 mM disodium EDTA, pH 7.8) and stored at −80 °C in small aliquots. Then the aptamer was dissolved in a nonfolding buffer consisting of HEPES binding buffer (HBB) (pH 7.4) supplemented with 1 mM TCEP and 3 mM of MgCl$_2$. All buffer solutions were prepared using a Milli-Q water purification system ($\geq$18 MΩ cm; Milli-Q, Millipore, Bedford, MA, USA) and used in all experiments. Screen-printed gold electrode assemblies (DRP-220AT) containing gold working and counter electrodes and Ag/AgCl reference electrodes were obtained from (DropSens, Metrohm, UK Ltd.). Differential pulse voltammetry (DPV) measurements were carried out using the potentiostat STAT8000P from (DropSens, Metrohm, UK Ltd.). A Bruker MultiMode 8-HR Atomic Force Microscopy (AFM) from (BRUKER, USA) was used for the visualisation of molecular layers deposited on the SPGE surface. AFM measurements were carried out at room temperature on one contact mode. NanoScope Analysis software (version 1.40) was used for the analysis of AFM images.

### 2.2. Nucleic-Acid-Based Sensor Development

#### 2.2.1. Fabrication of the Sensor

The aptamers were immobilised on the surface of gold screen-printed electrodes by the following procedure, shown schematically in Figure 1b. Before modification, gold electrodes were pretreated using the process detailed in our previous publications [26,37,39] with minor modifications. In this work, we used TCEP instead of 1,4-dithiothreitol (DTT) to break disulfide bonds in aptamers. Highly reactive thiol-substituted aptamers are typically supplied as aptamer dimers joined by the S-S bridge. Before immobilisation, the disulfide bridge should be split using Tris (2-carboxyethyl) phosphine hydrochloride (TCEP) as a reducing agent, which releases two aptamers with -SH groups at the 3′ terminal. In this work, a thiolated RNA-based aptamer (10 μM) was treated with freshly prepared 1 mM TCEP and 3 mM MgCl$_2$ in HEPES binding buffer (HBB) (pH 7.4) for 1 h at room temperature to cleave the disulfide bonds. This procedure was carried out for 1 h in darkness. Before immobilisation, the aptamer samples were activated in a thermocycler (Prime TC3600) by rapid heating to 95 °C for 1 min, followed by 1 min cooling to 4 °C. Then aptamer immobilisation on SPGE was performed by drop casting (20 μL) of 3 μM CG-3 aptamer solution and incubating samples for 12 h at 4 °C in a humidity chamber as previously described [37,38]. Then the electrodes were rinsed in HBB buffer. After immobilisation, the electrodes were ready to be used in PCA3 electrochemical measurements; alternatively, they can be stored at 4 °C in HBB to prevent aptamers from coiling.

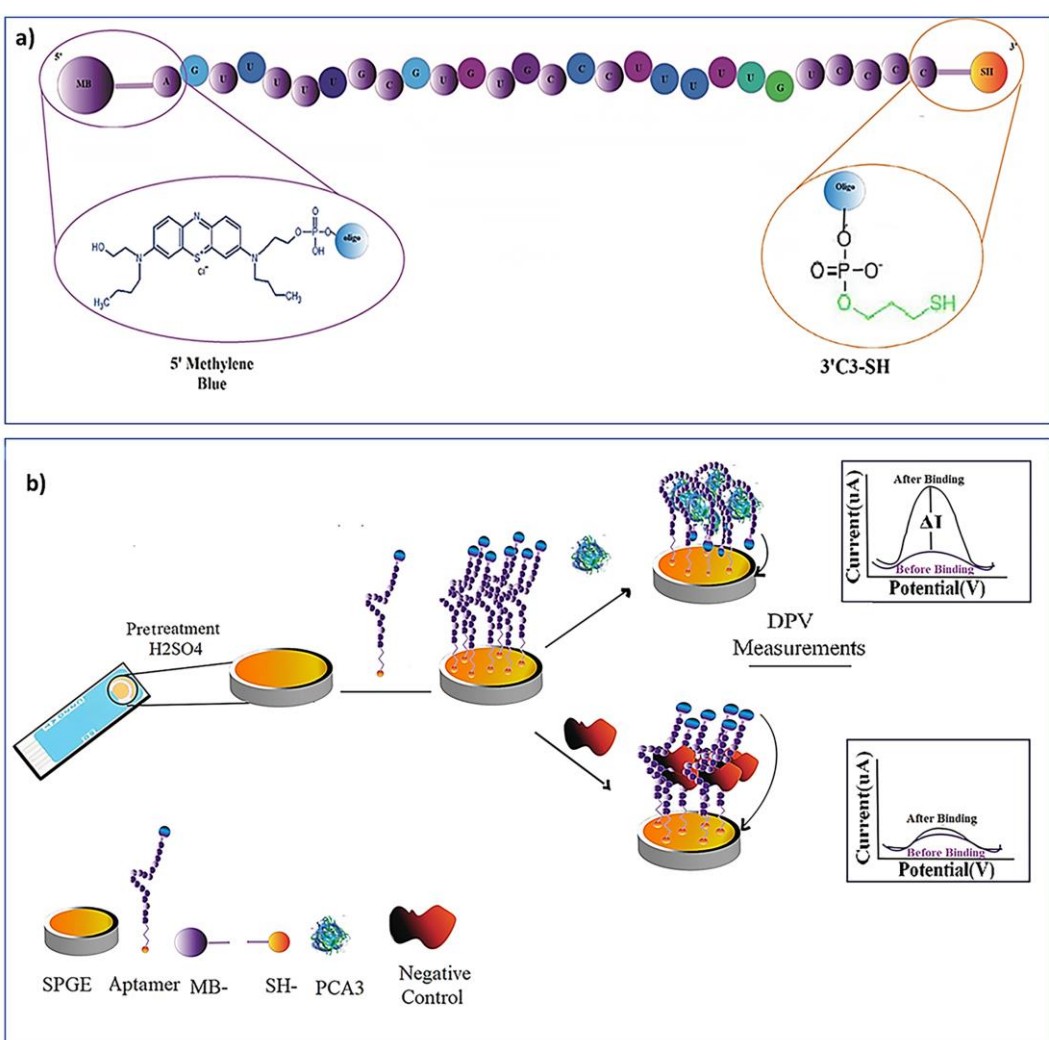

**Figure 1.** (**a**) Chemical structure of RNA aptamer specific to PCA3, including the functional groups of SH at the 3′ terminal and methylene blue at the 5′ terminal. (**b**) The fabrication and detection steps of a DPV aptasensor.

### 2.2.2. Electrochemical Detection of PCA3

The strategy of electrochemical DPV measurements, as outlined in Figure 1b, shows the electrodes with immobilised aptamers exposed to either the target analyte or the negative control analytes; the anticipated outcomes of DPV measurements were shown at the ends of the above two routes. For electrochemical measurements, the electrodes with immobilised aptamers were exposed to a small amount (40 μL); detection buffer (0.1 M PBS pH 7.5, 120 mM NaCl, 5 mM KCl) containing different concentrations of PCA3 (between 0.1 pM and 10 nM) was analysed on the modified aptasensor to create the calibration curve. These measurements were carried out in two ways: (i) by exposing a single aptasensor to solutions containing different PCA3 concentrations in sequential order starting from the lowest concentration of PCA3, and (ii) by using a new aptasensor for every PCA3 concentration. Before the electrochemical measurements, the electrodes with immobilised aptamers were incubated for 15 min in buffer solutions containing different concentrations of PCA3.

Differential pulse voltammetry (DPV) measurements were carried out in the potential range of −0.5 and −0.1 V at a scan rate of 0.02 V/s, a modulation amplitude of 0.002 V, a potential step of 0.004 V, and a pulse time of 10 ms. DPV measurements were carried out on electrodes exposed to different concentrations of PCA3. The analytical signal was the height of the resulting peak at roughly −0.4 V, which corresponded to the potential

redox range of the MB linked to the aptamer [44] (see Figure S1 in the supplementary material section). The aptasensor selectivity was examined by 15 min exposure to several negative controls, namely, scrambled PCA3 in several concentrations (0.1 nM, 10 nM, 100 nM, and 1 µM), PSA (10 nM), and BSA (10 nM). In addition to DPV, the aptamer to target (PCA3) affinity was evaluated from the amperometry kinetics study, that is, the time dependencies of cathodic current at −0.25 V recorded on electrodes during exposure to PCA3 of different concentrations as we described in our previous work [26,38]. The aptasensor performance was tested using samples based on synthetic urine (Surine™ negative control). Synthetic urine, a nonbiological combination often used as a negative reference standard in biosensing experiments [45–47], was used here as a medium instead of a buffer. Synthetic urine samples were spiked with PCA3 of different concentrations (from 0.1 pM to 10 nM) and used in DPV measurements with the same buffer protocols.

## 3. Results and Discussion

### 3.1. DPV Detection of PCA3 in Buffer

As shown in our earlier publication [39], the detection of PCA3 using cyclic voltammetry (CV) measurements on electrodes functionalised with a methylene-blue-labelled RNA aptamer demonstrated somewhat poor resolution. However, it contributed just to confirming the detection concept. In those measurements, the methylene blue characteristic peaks were invisible in pure buffer solution but started appearing when adding PCA3 to the buffer. The minimal detected concentration of PCA3 was 0.05 nM, close to the previously reported LDL of 0.1 nM for CV measurements with a ferrocene-labelled aptamer [37]. It seems that this is a limit for conventional CV measurements. Therefore, we decided to switch to a much more sensitive technique of digital pulse voltammetry (DPV). These measurements were conducted on gold screen-printed three-electrode assemblies (DropSens) with a methylene blue (MB)–labelled aptamer immobilised on the electrode surface in a buffer solution containing PCA3 in a wide range of concentrations from 0.1 pM to 10 nM. Typical results in Figure 2 show a correlation between the oxidation peak height and the concentration of PCA3 in the solution.

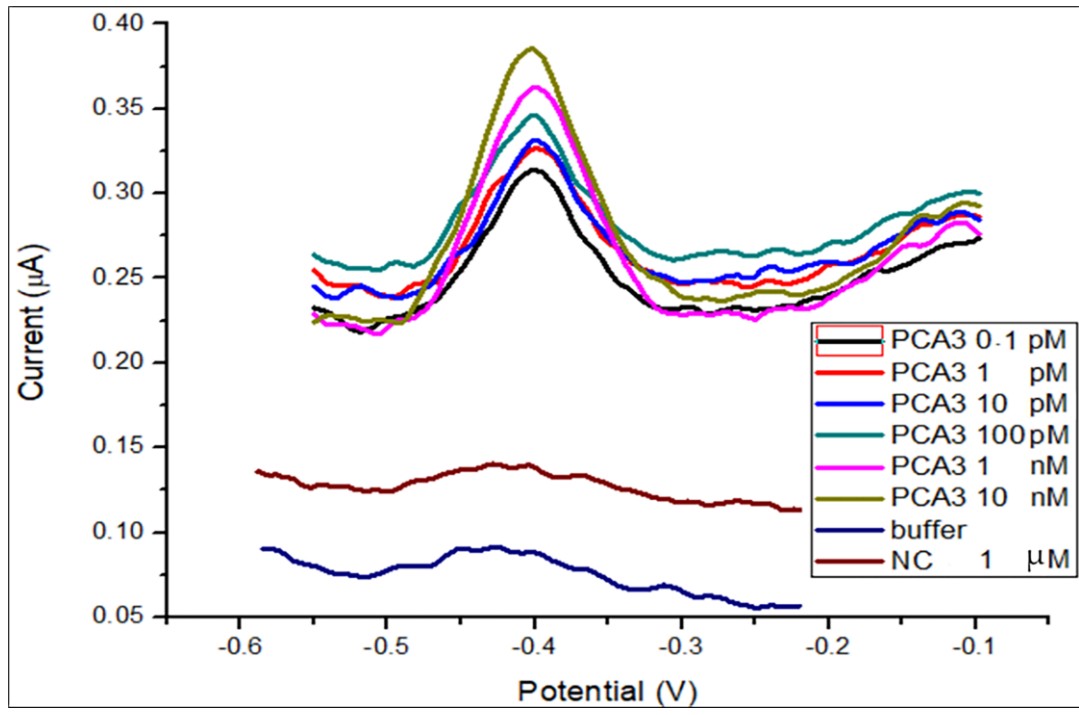

**Figure 2.** DPVs were recorded on electrodes functionalised with an MB-labelled aptamer in buffer solution containing different concentrations of PCA3. DPVs in pure buffer solution and negative control using scrambled PCA3 are shown for comparison.

Methylene blue (MB), a thiazine dye with a potential range between −0.1 and −0.4 V (vs. SCE) in pH 4–11 solution, is used as an electron transfer mediator because it is close to the redox potentials of several biomolecules [48]. As a result, the analytical signal was determined as the peak height at roughly −0.4 V, produced by the redox reaction of the MB linked to the aptamer as it approached the gold surface [44,49]. Methylene blue's oxidation signals were also identified using CV, as illustrated in the supplemental papers Figure S1. The curves of much lower intensities correspond to DPVs in buffer solution only and negative control scans in buffer containing PCA3 with a scrambled sequence of nucleotides. In addition, two other negative control tests were performed using BSA and PSA 10 nM solutions in the buffer. The results of all DPV measurements in buffer solution are summarised in Figure 3.

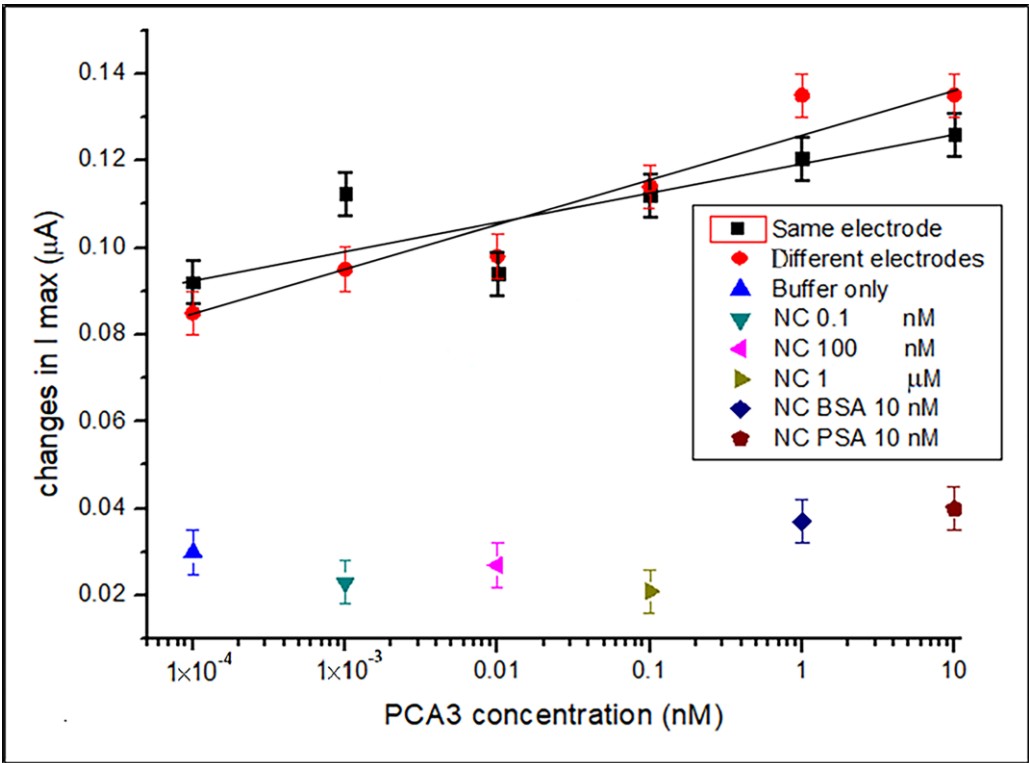

**Figure 3.** Dependences of the DPV peak current values vs. concentration of PCA3 in buffer solutions: black squares and line correspond to DPV measurements on the same electrode with PCA3 added sequentially; red circles and line correspond to DPV recorded on different electrodes. Individual data points below correspond to negative control tests using scrambled PCA3, BSA, and PSA.

As one can see, the DPV sensor response correlates almost linearly with the concentration of PCA3. Two DPV detection methods were carried out: (i) on the same electrode with sequential exposure to PCA3 of different concentrations starting from the smallest concentration and (ii) on different electrodes exposed to different concentrations of PCA3. Both series gave slightly similar results, which can justify using sequential measurements on a single electrode. The background level of a pure buffer of about 0.03 µA provides the reference point above the noise level of 0.005 µA. Therefore, an LDL of 0.1 pM could be evaluated as the response to the lowest concentration used, which is above the triple level of noise of 0.015 µA. Potentially, LDL could be even lower, but even the value of 0.1 pM is the lowest LDL reported for the electrochemical detection of PCA3, which could be sufficient for the detection of traces of PCA3 in regular urine tests. Furthermore, the data points for negative controls were very close to the background level, demonstrating high selectivity of PCA3 detection.

### 3.2. DPV Detection of PCA3 in Synthetic Urine

As a further step to the practical implementation of electrochemical aptasensing for PCA3, DPV measurements similar to those described in Section 3.1 were carried out on samples of synthetic urine spiked with PCA3 of different concentrations from 0.1 pM to 10 nM. The typical results are presented in Figure 4. Despite the higher noise level compared with DPV measurements in the buffer, a correlation between the DPV anodic peak current and the concentration of PCA3 is apparent. The above results and the data for a background level in pure urine and negative controls using scrambled PCA3, PBS, and PSA are summarised in Figure 5.

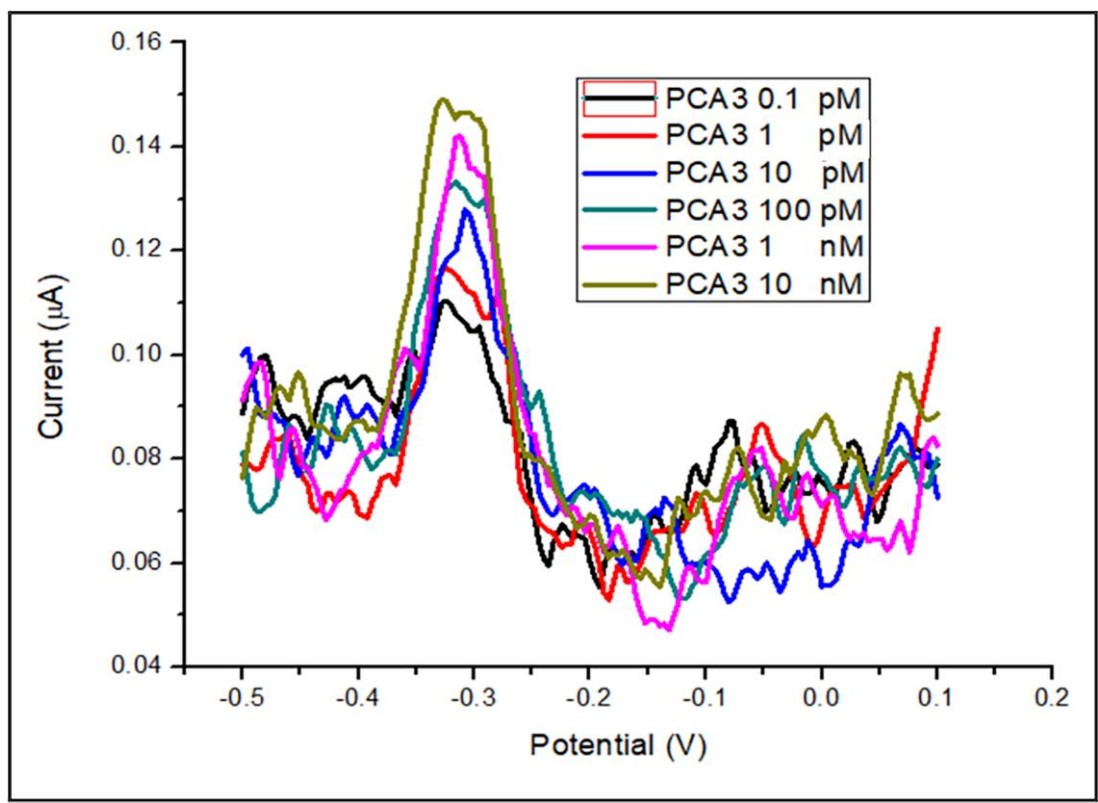

**Figure 4.** DPVs recorded on electrodes functionalised with an MB-labelled aptamer in synthetic urine containing different concentrations of PCA3.

Similar to the DPV data recorded in buffer solutions in Section 3.1, the results of DPV measurements in urine showed an almost linear correlation with the concentration of PCA3 for both single electrodes and different electrode types of measurements. The lowest concentration of used PCA3 of 0.1 pM, which is sufficiently above the triple noise level (0.015 μA), can be taken as LDL; however, it could potentially be lower. The background level in pure urine is about 0.097 μA, while negative controls in scrambled PCA3, BSA, and PSA are slightly higher than the background level but still lower than the responses to PCA3. The selectivity is still reasonably good, though a bit lower than that in buffer solution due to the interference of other proteins present in synthetic urine.

### 3.3. Aptamer/PCA3 Binding Kinetics Study

The affinity of the aptamer towards its target (PCA3) was evaluated from the study of binding kinetics using amperometry at a fixed potential. A typical series of time dependencies of current at a fixed potential of −0.25 V at different concentrations of PCA3 in buffer solution is shown in Figure 6. These measurements were carried out on the same electrode with PCA3 solutions added sequentially, starting from the smallest concentration of PCA3.

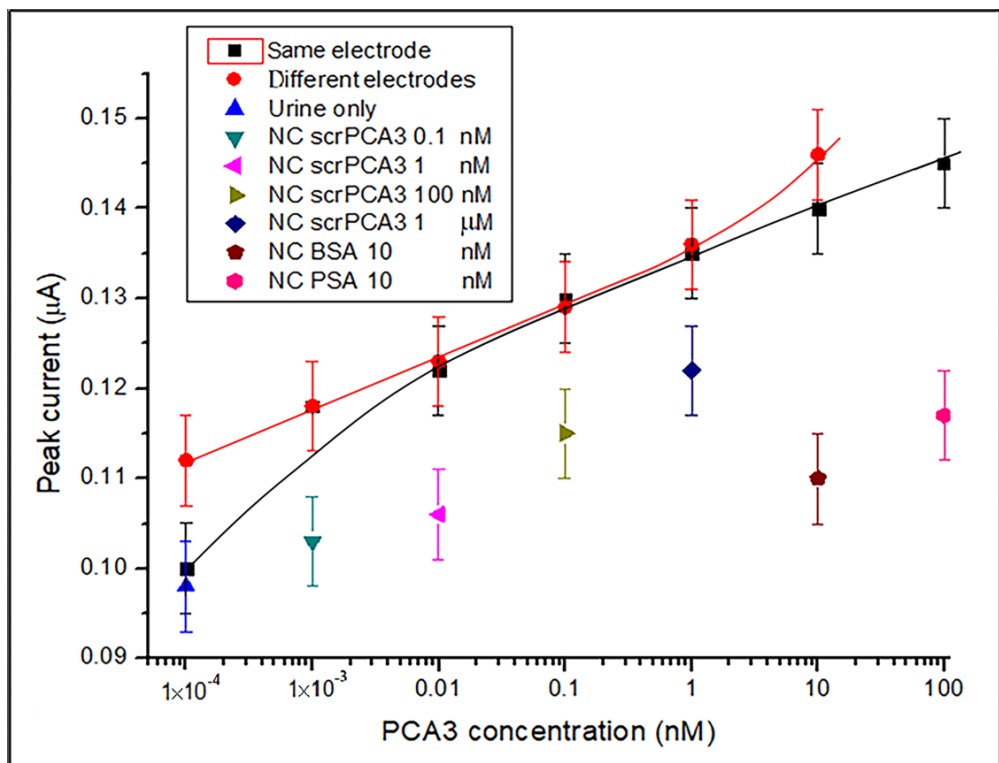

**Figure 5.** Dependences of the DPV peak current values vs. concentration of PCA3 in synthetic urine: black squares and line correspond to DPV measurements on the same electrode with PCA3 added sequentially; red circles and line correspond to DPV recorded on different electrodes. Individual data points below correspond to negative control tests using scrambled PCA3, BSA, and PSA.

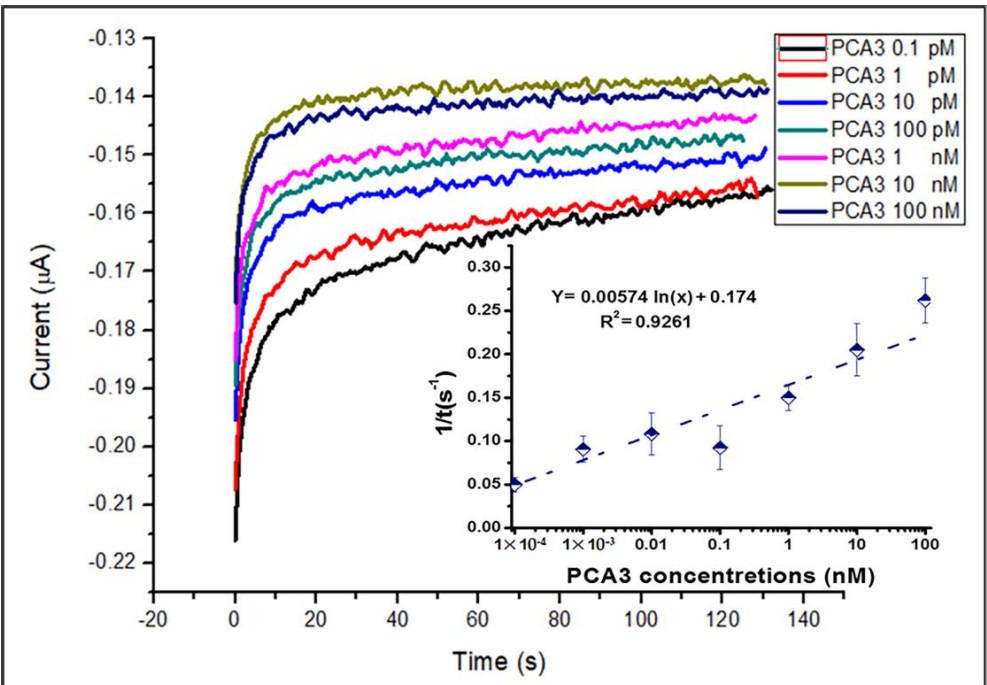

**Figure 6.** The time dependencies of current at −0.25 V were recorded at different concentrations of PCA3 in buffer solution. Inset shows the dependence of reciprocal time constant ($1/\tau$) against the concentration of PCA3.

Our earlier articles included examining binding kinetics and determining association and affinity constants [37,38]. Briefly, the time constants ($\tau$) were evaluated by fitting the kinetics curves to the rising exponential function. The reciprocal values ($1/\tau$) were plotted against the concentration of PCA3 (C), shown as an inset in Figure 6. The gradient of this graph ($1/\tau = k_a C + k_d$) gives the values of the adsorption ($k_a$) and desorption rates ($k_d$). The association constant is the ratio of adsorption and desorption rates $K_A = k_a/k_d$, while the affinity constant is its reciprocal $K_D = 1/K_A$. The parameters obtained from the graph (Figure 6) are: $k_a = 5.74 \times 10^{-3}$ ($\text{nM}^{-1}\text{s}^{-1}$) and $k_d = 0.1732$ (1/s), which yield $K_A = 3.3 \times 10^{7}$ ($\text{M}^{-1}$) and $K_D = 3.01 \times 10^{-8}$ (M). The obtained values confirm the high affinity of an RNA aptamer labelled with MB to its target PCA3, and these values are very close to those obtained earlier for DNA aptamers labelled with ferrocene [37] and unlabelled RNA aptamers [38].

### 3.4. AFM Study of PCA3 t Aptamer Binding

As shown earlier [48–50], AFM can provide valuable information on the morphology of the molecular layers deposited on the electrode surfaces in biosensors and also give evidence of aptamer/target binding. In this work, AFM measurements were carried out on the gold working electrode of DropSens screen-printed three-electrode assemblies using a Bruker MultiMode 8-HR AFM instrument. First, the aptamers were immobilised on the surface of gold following the procedure described in Section 2.2, and then the samples were incubated for 15 min in BBS containing 10 nM of PCA3. Finally, the tapping mode AFM measurements were performed on three samples: (a) bare gold electrode, (b) after immobilisation of aptamers, and (c) after binding PCA3. The results are shown in Figure 7 as 2D and pseudo-3D images.

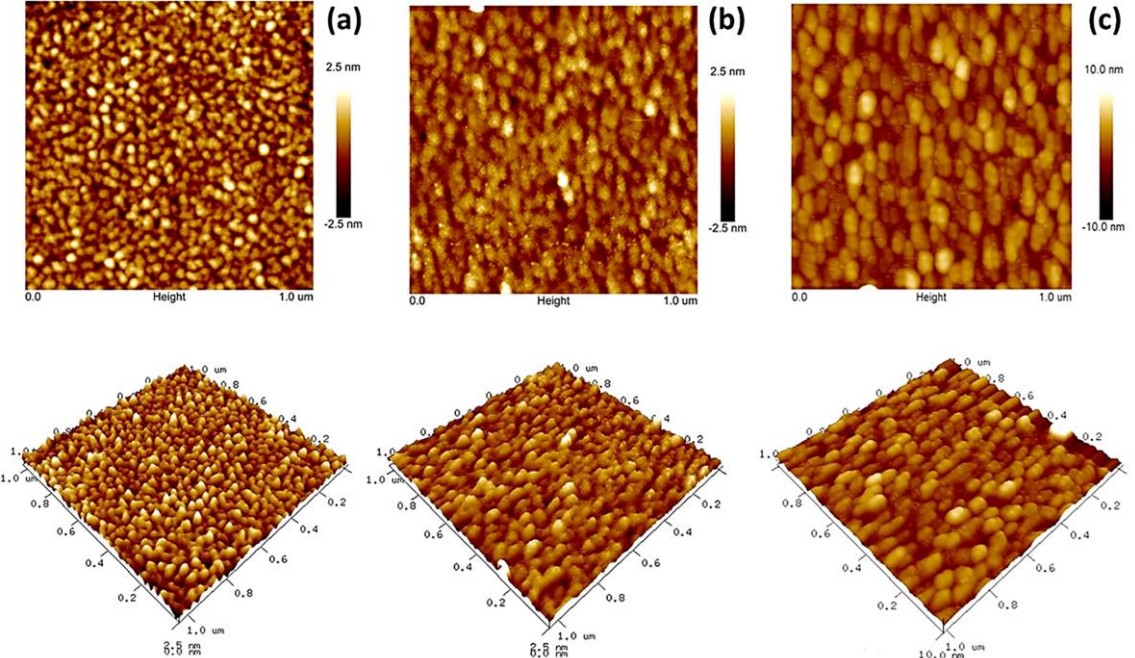

**Figure 7.** 2D and pseudo-3D AFM images of a bare gold electrode (**a**), after immobilisation of the aptamer (**b**), and after binding PCA3 to the aptamer (**c**).

The above AFM images show the typical grainy structure of gold. The presence of very thin (few nanometres) aptamer layers on the surface of gold did not make a dramatic difference to the AFM image. At the same time, the binding of a large RNA molecule of PCA3 causes a noticeable increase in grain sizes. Roughness analysis of 2D AFM images carried out using the NanoScope software (version 1.40) provided numerical evidence of PCA3-to-aptamer binding, as shown in Table 1. The values of average ($R_a$), root mean

square ($R_z$) and maximal roughness ($R_{max}$) were slightly decreased after the immobilisation of aptamers but increased substantially after biding PCA3 molecules to aptamers.

**Table 1.** The roughness parameters were evaluated on 2D AFM images in Figure 7.

| Samples | Roughness Parameters | | |
|:---:|:---:|:---:|:---:|
| | Ra (nm) | Rz (nm) | Rmax (nm) |
| (a) | 0.85 | 2.52 | 3.10 |
| (b) | 0.58 | 1.53 | 2.85 |
| (c) | 2.12 | 5.20 | 5.60 |

## 4. Conclusions

The exploitation of the DPV method in developing an electrochemical aptasensor for the detection of PCA3 was a success. The key point is that the detection of PCA3 in both buffer solutions and synthetic urine in a wide range of concentrations from 0.1 pM to 10 nM showed a linear calibration and an LDL of 0.1 pM. The ability to detect such low concentrations of PCA3 could be crucial for implementing this method for testing real samples of urine of prostate cancer patients. The excellent selectivity of PCA3 detection in buffer was confirmed by negative control tests using several analytes, namely, PCA3 with scrambled nucleotide sequences, BSA, and PSA. The responses to negative control analytes in high concentrations (10 nM of BSA and PSA and even 1 μM of scrambled PCA3) are smaller than the response to PCA3 in the lowest concentration of 0.1 pM. The reason for such excellent selectivity is the high affinity of the RNA aptamer to its specific target PCA3. The affinity constant of aptamer/PCA3 binding was evaluated as 0.3 μM from the binding kinetics study; in other words, this means that the aptamer/PCA3 binding is very strong, with the probability of detaching PCA3 from the aptamer being 3 million times smaller than the probability of binding. Judging from the negative control tests, the selectivity of a DPV aptasensor for the detection of PCA3 in synthetic urine is still good, although a bit lower than in buffer solutions. The interference and possible nonspecific binding of other proteins in synthetic urine is the reason for the decreased selectivity. This fact outlines challenges in future research and development of electrochemical aptasensor for detecting the prostate cancer biomarker PCA3 in urine samples of prostate cancer patients.

**Supplementary Materials:** The following supporting information can be downloaded at: https://www.mdpi.com/article/10.3390/eng4010022/s1, Figure S1: Electrochemical detection of PCA3 using cyclic voltammetry method.

**Author Contributions:** Conceptualisation, A.N.; methodology, A.N., D.S. and S.T.; investigation, S.T., A.L. and M.H.M.; data curation, S.T. and A.N., writing, S.T., M.H.M. and A.N., reviewing, D.S.; editing S.T. and M.H.M.; supervision, A.N., A.L. and D.S. All authors have read and agreed to the published version of the manuscript.

**Funding:** This research received no external funding.

**Institutional Review Board Statement:** Not applicable.

**Informed Consent Statement:** Not applicable.

**Data Availability Statement:** The data are not publicly available. The data files are stored in corresponding instruments in personal computers.

**Acknowledgments:** We would like to acknowledge Sheffield Hallam University, UK, specifically the Material and Engineering Research Institute (MERI) and the Biomolecular Sciences Research Centre (BMRC), City Campus, Howard Street, Sheffield, S1 1WB, for the full access to its resources and materials in this research.

**Conflicts of Interest:** The authors declare no conflict of interest.

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
