# Peer review of "Enhanced Performance Electrochemical Biosensor for Detection of Prostate Cancer Biomarker PCA3 Using Specific Aptamer"

_2673-4117, doi:10.3390/eng4010022_

Round 1

Reviewer 1 Report

Reviewer Recommendation and Comments for manuscript eng-2098344 with the title: “Enhanced Performance Electrochemical Biosensor for Detection of Prostate Cancer Biomarker PCA3 Using Specific Aptamer”, authors: S. Takita, A. Nabok, A. Lishchuk, M.H. Mussa, D. Smith.

The authors report obtaining an electrochemical biosensor based on screen-printed gold electrodes in combination with a methylene blue-labeled RNA-based aptamer specific for the detection of the prostate cancer biomarker PCA3.

The article may be published after revision.

The main comments that I find useful for improving the quality of the article are presented below:

*L176. “in our previous work [45].” Reference [45] must be checked. / [44] All references must be proofread for accuracy.

*L168-170. “The height of the resulting peak at around -0.4V, corresponding to the oxidation of MB attached to aptamer was taken as the analytical signal.” References or other scientific evidence should be added.

*3.1. DPV Detection of PCA3 In Buffer. DPV for buffer and control also show a small oxidation peak at -0.4 V. What are the corresponding oxidation processes?

*L254. “at a fixed potential of (0.25V) at different concentrations”. If the analytical signal is recorded at -0.4 V, why was such a high potential value used?

*L255. “These measurements were carried out on the same electrode”. For 140 sec. at 0.25 V, MB is oxidized at the electrode surface. How can the same electrode be reused? Is the electrode oxidized or not? New explanations need to be added!

*The insert in Figure 6. This is not a correct linearization. The root mean square deviation must be calculated. The values are far from linear!

*3.3. Aptamer/PCA3 Binding Kinetics Study. How was τ evaluated?

*Was a standard artificial urine protocol used to mimic natural human urine?

*The methodology for manufacturing new sensors involves several experiments/multiplications, evaluation of reproducibility, sensitivity, specificity, selectivity, degree of recovery, stability over time, conditions of use, storage, handling, etc. Perhaps a new section should be introduced to cover these aspects. Otherwise, the implementation of sensor is highly hazardous.

*The typos must be corrected.

L34. [5–7] .

L70. nanoparticles[23,24].

L146. 4°Cin

L156. 1uM

L157. 120 mm NaCl, 5 mm KCl

L164. aptasensor. .

L166. -0.1V … 0.02V/s, … 0.002V

L180. 0.1pM

etc.

*The eng journal require a specific format of references, authors must pay more attention in their writing. (e.g. et al.)

*There are some grammar and typing mistakes.

*The authors must revise the entire manuscript.

Author Response

Dear Editor / Reviewer

The authors were grateful for the reviewer's comments and suggestions. We tried our best to address all the critical comments, and the corrections were highlighted in yellow in the revised manuscript. The answers to the reviewer's comments are given below.

Answers to the first reviewer's comments

*L176. "in our previous work [45]." Reference [45] must be checked. / [44] All references must be proofread for accuracy. 

Reply: Yes, we agree. It was a reference system problem caused by transferring documents from the original article to the journal format documents. However, all references were checked, updated to the correct format and order, and updated in the revised manuscript.

*L168-170. "The height of the resulting peak at around -0.4V, corresponding to the oxidation of MB attached to aptamer was taken as the analytical signal." References or other scientific evidence should be added.?

 Reply: We will answer this question by including our CV measurement in supporting documents and with some references within the manuscript, as the methylene blue range typically ranges from -0.1 to -0.5 in some cases, reflecting the effect of biomolecular electronegativity and gold surface cleaning and activation.

*3.1. DPV Detection of PCA3 In Buffer. DPV for buffer and control also show a small oxidation peak at -0.4 V. What are the corresponding oxidation processes? 

Reply: Methylene blue (MB), a thiazine dye with a formal potential in the range of -0.1 to -0.4 V (vs SCE) in pH 4 to 11 medium (references were provided), is used as an electron transfer mediator because it is close to the redox potentials of many biomolecules. As a result, these signals may have appeared for the following reasons: We do not have redox chemicals in the solution, as stated clearly in the publication, but the redox group of MB is connected to the aptamer as a label (3uM). As a result, in the absence of PCA3 in the solution, aptamers stretch, and the redox label (MB) moves away from the surface, resulting in limited charge transfer and, as a result, low oxidation and reduction currents. When PCA3 is added to the solution, the aptamers bind their targets (PCA3) by wrapping around, bringing the redox label closer to the electrode surface and increasing charge transfer. The NC was used at high concentrations (up to 1uM) during the NC study and still had a conductivity effect. However, compared to signals containing PCA3, these signals are considered insignificant.

*L254. “at a fixed potential of (0.25V) at different concentrations”. If the analytical signal is recorded at -0.4 V, why was such a high potential value used? 

Reply: Yes, the reviewer is right; It was a mistake (missing the negative sign) in the paper. The fixed potential -0.25V was used for the kinetic measurements.   

*L255. "These measurements were carried out on the same electrode". For 140 sec. at 0.25 V, MB is oxidised at the electrode surface. How can the same electrode be reused? Is the electrode oxidised or not? New explanations need to be added!

 Reply: Yes, the measurements were carried out at -0.25; this typo error is corrected in the revised manuscript. However, gold electrodes are neither oxidised nor reduced. As we explained earlier, the redox current is due to the charge transfer between the redox (MB) label and the electrode. Following the Langmuir model of molecular adsorption, which is applicable in our case of aptamer monolayers immobilised on the surface, the same electrode can be used by sequentially exposing it to solutions containing varying concentrations of PCA3, starting with the lowest concentrations; this allows for a decrease in the number of electrodes used as part of the platform optimisation. However, for DPV measurements, we discovered that using fresh/new electrodes for each concentration of PCA3 gives more precise and reliable findings than using a single electrode. The process was described in detail in the publication.

*The insert in Figure 6. This is not a correct linearisation. The root means square deviation must be calculated. The values are far from linear! 

Reply: - The values 1/t are derived from a series of measurements shown in fig. 6 and have been modified. Also, the root means square deviation was calculated at the acceptable range of about 0.92% at the exponential fittings. So, yes, the data points deviate from the straight line, but at least show the trend of a monotonic increase of 1/t with PCA3 concentration.

*3.3. Aptamer/PCA3 Binding Kinetics Study. How was τ evaluated?

Reply: The time constant values (t) were evaluated for every concentration of PCA3, fitting the data shown in Fig. 6 to the rising exponential function 1-exp(-t/t), as described previously.

 *Was a standard artificial urine protocol used to mimic natural human urine?  

Reply: confirmative; according to Sigma Aldrich, the formula is standard with all urine components to mimic human urine, as we supported with other references. Using real urine samples from PCa patients was not planned for this study, but it will be in the future.

*The methodology for manufacturing new sensors involves several experiments/multiplications, evaluation of reproducibility, sensitivity, specificity, selectivity, degree of recovery, stability over time, conditions of use, storage, handling, etc. Perhaps a new section should be introduced to cover these aspects. Otherwise, the implementation of sensor is highly hazardous.

Reply: We reported on the work done, which is clearly far from the sensor implementation stage. Sensitivity and specificity/selectivity were investigated. Sensor recovery was investigated, but the results were not provided in this work because the recovery was incomplete; therefore, the described sensors are single-use. Other aspects, such as long-term stability and storage, have not yet been investigated yet. We appreciate the reviewer's suggestions and will address these issues in our future work.

*The typos must be corrected.

All typos are corrected in the updated manuscript; however, the journal has a strong and effective editing and checking system for these mistakes before it is published.

*The eng journal require a specific format of references, authors must pay more attention in their writing. (e.g. et al.)

 Concerning this comment, the MDPI journals in applied and engineering disciplines adopted a numbering reference in any acceptable format as mentioned in the instructions for authors ( https://www.mdpi.com/journal/eng/instructions  ), and the reference format used here is IEEE format

 "References: References must be numbered in order of appearance in the text (including table captions and figure legends) and listed individually at the end of the manuscript. We recommend preparing the references with a bibliography software package, such as EndNote, Reference Manager or Zotero, to avoid typing mistakes and duplicated references. We encourage citations to data, computer code and other citable research material. If available online, you may use reference style 9. Below".

*Extensive editing of English language and style required

A native English speaker proofread the English, and the errors were corrected. Also, the layout and general editing are corrected according to MDPI manuscript gaudiness

Finally, the authors would like to thank the reviewer for enhancing the clarity and the scientific quality by addressing his/her comment on this paper

With kind regards

Sarra Takita

The corresponding author

Reviewer 2 Report

I have read with interest the manuscript by Sarra Takita et al. entitled “Enhanced Performance Electrochemical Biosensor for Detection of Prostate Cancer Biomarker PCA3 Using Specific Aptamer”. While I appreciate the intent of the work there are several concerns and suggestions that the authors should address to prepare the MS for publication.

1) Section 2.1 Reagents and apparatus Lack the information about sequences of oligonucleotides used in experiments and their purity. In the case GC3 aptamer please add information about thiol and methylene blue modifiers. 

2)    Section 2.2.1 Fabrication of The Sensor 

Breaking of disulfide bonds should be described in detail (lack the information about TECP concentration, pH of working TECP solution).   

3)    Section 2.2.2. Electrochemical Detection of PCA3

The author should add information about the sensor regeneration procedure during analysis of two samples. 

4) Fig.2 Legend with concentrations should be corrected. 

5)  Did the author perform an analysis of samples with a concentration lower than 100 fM such as 10 and 1 fM? Differences between the negative sample and the sample with 100 fM suggest a lower detection limit. Please provide results. 

6) Experimental Raw data (especially form electrochemical measurements) should be provided as supplementary data in csv or txt format.

Author Response

Dear Editor / Reviewer

The authors were grateful for the reviewer's comments and suggestions. We tried our best to address all the critical comments, and the corrections were highlighted in yellow in the revised manuscript. The answers to the reviewer's comments are given below.

Answers to the Second reviewer's comments

1) Section 2.1 Reagents and apparatus Lack the information about sequences of oligonucleotides used in experiments and their purity. In the case GC3 aptamer please add information about thiol and methylene blue modifiers. 

Reply: This study builds on earlier work; we previously mentioned the aptamer sequences in references; the aptamer sequence was shown in the diagram in Fig. 1. However, based on your comment, we included it in the text of the revised manuscript. Functionalization of the aptamer by thiols at 3' and methylene Blue at 5' has been done by the company Merck Life Science Ltd. (Dorset, UK).

2)    Section 2.2.1 Fabrication of The Sensor 

Breaking of disulfide bonds should be described in detail (lack the information about TECP concentration, pH of working TECP solution).   

Reply: Highly reactive thiol-substituted aptamers are usually supplied as aptamer dimers joined by the S=S bridge. Before immobilization, the disulfide bridge will be split using TCEP, which releases two aptamers with SH groups at the C3' terminal. The information on TCEP concentration and pH values of the solution contained was provided in section 2.1, "Reagents and apparatus". Corresponding changes were added in the manuscript in section 2.2.1, "Fabrication of The Sensor."

3)    Section 2.2.2. Electrochemical Detection of PCA3

The author should add information about the sensor regeneration procedure during analysis of two samples. 

Reply: We did not use the sensor regeneration during the measurements on a single aptasensor. Following the Langmuir model of molecular adsorption (binding), which is applicable in our case of an aptamer monolayer on the electrode surface, the molecular adsorption can be carried out in sequential order starting from the smallest concentration of the analyte. The results obtained proved that both methods (e. g. the use of single and multiple electrodes) gave similar results. However, we observed that using several electrodes gives considerably more precise and reliable results than a single electrode. The process was detailed in detail in the publication. Aptasensor recovery was explored but not included in this study; the recovery was incomplete, and therefore, the proposed sensors are of single-use only.

 4) Fig.2 Legend with concentrations should be corrected. 

Reply: Yes, the concentrations in the legend will be updated in the amended manuscript's Fig2.

5)  Did the author perform an analysis of samples with a concentration lower than 100 fM, such as 10 and 1 fM? Differences between the negative sample and the sample with 100 fM suggest a lower detection limit. Please provide results. 

Reply: The lowest PCA3 concentration used was 0.1 pM gives a response which is about three times larger than the reference (pure buffer solution), so the LDL values are justified, as mentioned in work.

6) Experimental Raw data (especially form electrochemical measurements) should be provided as supplementary data in csv or txt format.

Reply: We provided the original data plotted using Origin software

Finally, the authors would thank the reviewer for enhancing the clarity and the scientific quality by addressing his/her comment on this paper

With kind regards

Sarra Takita

The corresponding author

Reviewer 3 Report

This study evaluates the DPV method to develop an aptasensor to detect the biomarker of prostate cancer PCA3. It is a very well-written work, with scientific rigour and a very well-applied methodology. The results are quite promising

Author Response

Dear Editor / Reviewer

The authors were grateful for the reviewer's comments and suggestions. We tried our best to address all the critical comments from all reviewers, and the corrections were highlighted in yellow in the revised manuscript. The answers to the reviewer's comments are given below.

Reviewer comments and answers

Comments and Suggestions for Authors

This study evaluates the DPV method to develop an aptasensor to detect the biomarker of prostate cancer PCA3. It is a very well-written work with scientific rigour and a very well-applied methodology. The results are quite promising

Reply: We appreciate the high estimate of our work by the reviewer.

With kind regards

Sarra Takita

The corresponding author

Round 2

Reviewer 1 Report

Reviewer Recommendation and Comments for manuscript eng-2098344 with the title: “Enhanced Performance Electrochemical Biosensor for Detection of Prostate Cancer Biomarker PCA3 Using Specific Aptamer”, authors: S. Takita, A. Nabok, A. Lishchuk, M.H. Mussa, D. Smith.

The authors provide a new improved version of the manuscript that takes into account the reviewers' comments. Technically, I have one more concern:

Line 16 - “digital pulse voltammetry (DPV)” What is the difference between digital pulse voltammetry (DPV) and differential pulse voltammetry (DPV)? Is it the same electrochemical method? Why digital?

Authors need to correct any grammatical/editing/typing errors much more carefully before publication.

Line 141 - [26], [37,39]

Line 154 - [37], [38]

Line 189 - [45]–[47]

If the authors consider it appropriate, Fig.S1 of the supplementary material can be inserted into the manuscript.

Line 194 – “Cyclic Voltammogram (CV) measurements” Cyclic Voltammetry (CV) measurements

Is PCa and PCA the same? If yes, then the same notation should be used.

Figures 7b and 7c are unclear.

*The eng journal require a specific format of references, authors must pay more attention in their writing. No reference is written according to the format required by the journal.

Author Response

Dear Editor / Reviewer

The authors were grateful for the reviewer's comments and suggestions. We tried our best to address all the critical comments, and the corrections of the second round were highlighted in green in the revised manuscript. The answers to the reviewer's comments are given below.

Answers to the first reviewer's comments:

Line 16 - “digital pulse voltammetry (DPV)” What is the difference between digital pulse voltammetry (DPV) and differential pulse voltammetry (DPV)? Is it the same electrochemical method? Why digital?

Reply: in response to your comment, we have changed this typo mistake into differential pulse voltammetry.

Authors need to correct any grammatical/editing/typing errors much more carefully before publication.

Line 141 - [26], [37,39]

Line 154 - [37], [38]

Line 189 - [45]–[47]

Reply: the mentioned reference has been adjusted as you suggested. However, we notice that these mistakes appear after MDPI formatted the manuscript.

If the authors consider it appropriate, Fig.S1 of the supplementary material can be inserted into the manuscript.

Reply: Thank you for your suggestion, but the primary focus of this work is on DPV measurement of PCA3 in buffer and urine, and the use of CV is just to determine the potential MB region. However, we appreciate your feedback and will take it into consideration in our future work.

Line 194 – “Cyclic Voltammogram (CV) measurements” Cyclic Voltammetry (CV) measurements

Reply: This typo mistake has been addressed.

Is PCa and PCA the same? If yes, then the same notation should be used.

Reply: PCa is an abbreviation for prostate cancer as mentioned in line 31, while PCA3 stands for prostate cancer antigen 3 which is a biomarker specific for PCa as it mentioned in line 57.

Figures 7b and 7c are unclear.

Reply: This feedback has been taken into consideration and adjusted.

*The eng journal require a specific format of references, authors must pay more attention in their writing. No reference is written according to the format required by the journal.

Reply: the mentioned reference has been adjusted as you suggested. However, we notice that these mistakes appear after MDPI formatted the manuscript.

Finally, the authors would like to thank the reviewer for enhancing the clarity and the scientific quality by addressing these comments on this paper.

With kind regards

Sarra Takita

The corresponding author

Reviewer 2 Report

Dear Authors, 

Please read my comments and add the required additional information to the manuscript:

1) Section 2.1 Reagents and apparatus Lack the information about sequences of oligonucleotides used in experiments and their purity. In the case GC3 aptamer please add information about thiol and methylene blue modifiers. 

Reply: This study builds on earlier work; we previously mentioned the aptamer sequences in references; the aptamer sequence was shown in the diagram in Fig. 1. However, based on your comment, we included it in the text of the revised manuscript. Functionalization of the aptamer by thiols at 3' and methylene Blue at 5' has been done by the company Merck Life Science Ltd. (Dorset, UK).

Thank you for your response.

Introduced information about oligonucleotides is not sufficient:

- lncRNA PCA3 and scrambled PCA3 sequences must be added,

- additional information about the thiol modifier must be added, please specify the name of the thiol modifier.  

2)    Section 2.2.1 Fabrication of The Sensor 

Breaking of disulfide bonds should be described in detail (lack the information about TECP concentration, pH of working TECP solution).   

Reply: Highly reactive thiol-substituted aptamers are usually supplied as aptamer dimers joined by the S=S bridge. Before immobilization, the disulfide bridge will be split using TCEP, which releases two aptamers with SH groups at the C3' terminal. The information on TCEP concentration and pH values of the solution contained was provided in section 2.1, "Reagents and apparatus"Corresponding changes were added in the manuscript in section 2.2.1, "Fabrication of The Sensor."

Lines 143-148 contain incorrect information, thiol modified nucleotides are provided in two options:

a)     with S-S bridge that merges oligonucleotide with a short carbon chain that must be cleaved by TCEP or other chemicals

b)     oligonucleotide with a free thiol group (that is stored under an argon atmosphere)

Please correct the information according to your oligonucleotide synthesis option. 

Author Response

Dear Editor / Reviewer

The authors were grateful for the reviewer's comments and suggestions. We tried our best to address all the critical comments, and the corrections of the second round were highlighted in green in the revised manuscript. The answers to the reviewer's comments are given below.

Answers to the second reviewer's comments:

Please read my comments and add the required additional information to the manuscript:

  • Section 2.1 Reagents and apparatus Lack the information about sequences of oligonucleotides used in experiments and their purity. In the case GC3 aptamer please add information about thiol and methylene blue modifiers. 

Reply: This study builds on earlier work; we previously mentioned the aptamer sequences in references; the aptamer sequence was shown in the diagram in Fig. 1. However, based on your comment, we included it in the text of the revised manuscript. Functionalization of the aptamer by thiols at 3' and methylene Blue at 5' has been done by the company Merck Life Science Ltd. (Dorset, UK).

Thank you for your response.

Introduced information about oligonucleotides is not sufficient:

- lncRNA PCA3 and scrambled PCA3 sequences must be added,

- additional information about the thiol modifier must be added, please specify the name of the thiol modifier.  

Reply: The manuscript's included sequence shows the reduced form after adding TCEP that is ready to be immobalised on the gold surface which slightly updated in green highlight. While the fragment of lncRNA PCA3-227 was determined from the GenBank database, both the chosen sequence and its scrambled form (negative control) were customised by Eurofins Genomics (Germany). This on-going research is part of an attempt to investigate the most sensitive and cost-effective method for determining PCA3. The chosen/optimised detection technique will then be applied to detect pca3 in real patients' urine.

  • Section 2.2.1 Fabrication of The Sensor 

Breaking of disulfide bonds should be described in detail (lack the information about TECP concentration, pH of working TECP solution).  

Reply: Highly reactive thiol-substituted aptamers are usually supplied as aptamer dimers joined by the S=S bridge. Before immobilization, the disulfide bridge will be split using TCEP, which releases two aptamers with SH groups at the C3' terminal. The information on TCEP concentration and pH values of the solution contained was provided in section 2.1, "Reagents and apparatus". Corresponding changes were added in the manuscript in section 2.2.1, "Fabrication of The Sensor."

Lines 143-148 contain incorrect information, thiol modified nucleotides are provided in two options:

  1. a)     with S-S bridge that merges oligonucleotide with a short carbon chain that must be cleaved by TCEP or other chemicals
  2. b)     oligonucleotide with a free thiol group (that is stored under an argon atmosphere)

Please correct the information according to your oligonucleotide synthesis option. 

Reply: We confirmed that the method's sequence showed the reduced form following the addition of TCEP. The stored oligo in other hand was stored in unreduced form, which was acquired from Merck Life Science Ltd. (Dorset, UK). The working aptamer was then reduced in a non-folding buffer made up of HEPES binding buffer (pH 7.4), 1 mM TCEP, and 3 mM MgCl2. As described in Section 2.1. Reagents and apparatus.

Finally, the authors would like to thank the reviewer for enhancing the clarity and the scientific quality by addressing these comments on this paper.

With kind regards

Sarra Takita

The corresponding author
